# The influence of physical exercise on emotional management ability in college students: A serial mediation model of exercise adherence and psychological resilience

Yu-han Li[1], Wei-dong Zhu[1☺], Hu Lou[1], Bo Li[1], Shan-shan Han 🄳[1]*, Yu-yan Qian[2]*

**1** Institute of Sports Science, Nantong University, Nantong, China, **2** Journal Publishing Center, Nantong University, Nantong, China

☺ These authors contributed equally to this work.
* paiqiuhanshanshan@163.com (S-sH); 105637993@qq.com (Y-yQ)

## Abstract

### Objective

This study aimed to examine the influence of physical exercise (PE) on emotional management ability (EMA) among college students and to investigate the mediating roles of exercise adherence (EA) and psychological resilience (PR) in the relationship between PE and EMA.

### Method

Demographic information was collected using stratified, cluster, and multi-stage sampling methods. Data on PE, EA, PR, and EMA were collected from college students via the Wenjuanxing platform, resulting in a final sample of 11,388 valid questionnaires.

### Results

Significant positive correlations were found between PE and EMA (r = 0.164, *P* < 0.01), PE and EA (r = 0.452, *P* < 0.01), and PE and PR (r = 0.201, *P* < 0.01). Significant positive correlations were also observed between EA and PR (r = 0.329, *P* < 0.01), and EA and EMA (r = 0.478, *P* < 0.01). PR was significantly positively correlated with EMA (r = 0.411, *P* < 0.01). PE significantly negatively predicted EMA (β = −0.009, *P* < 0.001). EA mediated the relationship between PE and EMA (95% CI: [0.039, 0.045]). PR also mediated the relationship between PE and EMA (95% CI: [0.001, 0.004]). A serial mediation effect of EA and PR was found in the relationship between PE and EMA (95% CI: [0.008, 0.010]).

**Data availability statement:** Due to ethical restrictions of disclosing personal and sensitive data in accordance with the protocol approved by the Ethics Review Board of Nantong University, authors have to seek permission to allow us to make the data used in this study available. Data will be available upon request after permission is granted from the Ethics Review Board of Nantong University. Inquiries for data access should be sent to the Ethical Review Authority, whose contact is kjc@ntu.edu.cn for permission to openly share the data.

**Funding:** This study was supported by the 2024 General Project of Philosophy and Social Science Research at Universities in Jiangsu Province. (NO: 2024SJYB1253).

**Competing interests:** The authors have declared that no competing interests exist.

**Abbreviations:** CPAHLS-CS, Chinese College Students Physical Activity and Health Longitudinal Survey; EA, Exercise Adherence; EMA, Emotional Management Ability; EIS, Emotional Intelligence Scale; HL, Health Literacy; HLS-SF9, Health Literacy Scale-Short Form 9; PARS-3, Physical Activity Rating Scale; PE, Physical Exercise; PR, Psychological Resilience.

## Conclusion

This study reveals the impact of PE, EA, and PR on EMA. The findings suggest that enhancing PE participation, improving EA, and increasing PR can improve EMA among college students.

## Introduction

Emotional management refers to an individual's ability to control and regulate their emotional state. With economic and social development, negative emotional experiences, such as anxiety and depression, have significantly increased in China [1]. College students facing pressures related to academic performance, the pursuit of success, and post-graduation planning experience higher rates of depression, anxiety, and stress [2]. Research indicates that anxiety, depression, and even suicidal ideation are becoming the most common mental health issues among Chinese college students [3]. The "2022 Survey Report on the Mental Health Status of College Students" revealed that 21.48% of Chinese college students may be at risk for depression, and 45.28% may be at risk for anxiety [4]. Long-term emotional distress can lead to problems such as sleep disorders and eating disorders [5]. Previous studies have explored the link between emotional problems (social anxiety) and mobile-phone addiction, pointing out that media addiction is associated with cognitive and attitudinal dysfunction. Specifically, social anxiety related to social-media use may ultimately result in poor emotional-management ability [6]. Individuals with poor emotional management abilities are more likely to experience anxiety, depression, and other issues, increasing the risk of suicidal ideation or behaviors [7], and causing irreparable harm to families, the nation, and society. Conversely, individuals with good emotional management abilities can promptly adjust negative emotions, maintain physical and mental health, actively cope with academic and work stress, improve academic performance, and increase the likelihood of successfully completing academic courses [8]. Emotion regulation ability is essential to mental health, guiding mental health development [9]. Current research suggests significant gender differences in emotional management ability (EMA), with females often demonstrating more extraordinary abilities than males. This may be attributed to lower female hormone secretion than males, as hormones influence aggressiveness and control impulses [10]. EMA has become a key area of research in modern society, attracting widespread academic attention. Enhancing EMA can effectively prevent and reduce mental health issues, promoting the overall development of college students and the stability of society. Prior research suggests that physical exercise (PE) helps develop personality traits, cognitive styles, and social support in young people and positively alleviates psychological distress [9]. Physical exercise promotes personality traits, cognitive styles, and social support among the youth, aiding in alleviating psychological distress. Regular aerobic exercise can activate the motor center of the cerebral cortex, replacing involuntary exercise with constructive voluntary exercise to address anxiety and adverse stress [11]. High levels of physical exercise can enhance

endothelial cell function, promote the release of pro- and anti-inflammatory cytokines, reduce the incidence of viral infections, and effectively lower suicide risk [12]. This reduction in psychological distress frees cognitive resources, enabling more adaptive emotion regulation and increasing positive affect, thereby ultimately enhancing emotional management ability. Therefore, considering the influencing factors of PE, we are combined with the understanding of EMA.

PE is globally recognized as one of the most important pathways to health promotion, with some countries incorporating it into national health policies as a crucial measure for promoting health development [13]. However, PE levels among adults worldwide are severely insufficient [14]. Previous research has found that PE is an effective method for treating psychological disorders and alleviating anxiety and depression [15]. Regular PE can induce hippocampal neurogenesis, reverse brain neurodegeneration, enhance neuroplasticity, and improve cognitive function [16]. Additionally, PE has critical social components, helping to meet adolescents' social needs and expectations, improve their interpersonal and social skills, and prevent loneliness [17,18]. Further studies have found that exercise can enhance and regulate the mental health of college students, release psychological stress to some extent, increase social opportunities through exercise, and reduce negative emotions such as loneliness [19]. PE contributes to the development of non-cognitive factors, enhancing self-confidence, responsibility, sense of honor, and collectivism, cultivating perseverance, decisiveness, self-control, independence, and other personality qualities, making individuals cheerful, optimistic, emotionally invigorated, and full of vitality, enhancing self-understanding, and fostering self-acceptance [20]. Therefore, the importance of PE will become increasingly prominent.

Exercise adherence (EA) refers to a long-term, regular PE behavior completed by an individual through emotional and volitional effort [21]. From the perspective of physiological changes, adhering to PE can continuously stimulate the body to secrete dopamine, which is closely related to pleasure, helping to enhance positive emotions [22]. However, some surveys indicate that a considerable number of college students experience withdrawal and avoidance during EA [23]. With the acceleration of modern life and increased work pressure, more and more people neglect PE and lack long-term exercise habits [24]. Related studies have shown that people who do not exercise for a long time experience a gradual decline in physical fitness and are prone to problems such as obesity, increased risk of cardiovascular disease, and weakened muscle strength. People who lack exercise often feel tired in daily activities, have poor mental states, and reduced work efficiency. As they age, the decline in physical function becomes more pronounced, severely affecting their overall quality of life [25,26]. Studies have shown that people who can adhere to exercise tend to have stronger self-discipline and willpower. They can not only maintain good physical conditions but also release stress, improve mood, and enhance psychological resilience (PR) during exercise, thereby better coping with various challenges in life. They also demonstrate higher efficiency and concentration in learning and work, further enhancing their comprehensive qualities and life satisfaction, laying a solid health foundation for personal growth and development [27]. Research also indicates that continuous exercise behavior increases the search, understanding, and application of health-related electronic information. This may be because individuals want to manage their health better or optimize their exercise plans. Increased Health literacy enables individuals to access and utilize health information more effectively, including strategies that may help regulate and manage emotions, thereby improving EMA; i.e., EA is positively correlated with EMA [28].

Psychological resilience (PR) refers to the process of adapting well in the face of adversity, trauma, tragedy, threats, or even significant sources of stress [29]. Currently, there are three main perspectives on the definition of PR in foreign research: PR is the positive outcome of an individual experiencing high risk, such as growth after experiencing highly dangerous or challenging events, reflecting positive transformation; PR is a dynamic process in which stressful, adverse life events and protective factors work together, such as individuals continuously adjusting and adapting when facing adverse life events such as stress and adversity, achieving the development of PR; PR is an individual's ability or trait to cope with adverse life events such as stress, frustration, and trauma, such as rapidly adjusting after experiencing failure and actively recreating. According to existing research, PR includes five dimensions: goal focus, interpersonal assistance, family support, emotional control, and positive cognition [30]. Literature research has found that

researchers increasingly recognize the view of PR as a trait or ability [31]. PR is considered an essential psychological quality related to an individual's health and well-being, and everyone's PR can be improved and enhanced through practical training [32]. In this study, we conceptualize PR as both a dynamic process and a trait. As a dynamic process, PR involves the continuous adjustment and adaptation to stressful events, reflecting the interplay between risk and protective factors. As a trait, PR represents an individual's inherent capacity to withstand and recover from adversity, which can be enhanced through experiences and training. Survey results show that the physical health status of college students in recent years is not optimistic, their interest in sports is not high, especially the lack of rule awareness, competitive awareness, and teamwork awareness inherent in the spirit of sports, and the ability of college students to withstand setbacks and resist stress shows a weakening trend [33]. In China, college students generally have weak stress resistance and poor psychological endurance, leading them to have thoughts of ending their lives when facing stress and setbacks, and are prone to extreme behaviors such as suicide and self-harm [34], showing the cumulative adverse effects caused by fluctuations in the social mentality of college students [35]. According to many empirical studies, resilience is negatively correlated with mental health indicators (such as depression, anxiety, and negative emotions) and positively correlated with positive mental health indicators (such as life satisfaction, subjective well-being, and positive emotions) [36]. The total score of PR is also positively correlated with adaptability, emotional management, and positive cognition, indicating that the improvement of PR among college students can improve good adaptability and emotional regulation ability and strengthen positive cognitive attitudes [37]. Experimental evidence shows that highly resilient participants immediately adopt strategies such as "positive reappraisal + humor" after failure-induced stress, and their cardiovascular recovery is significantly faster than that of the low-resilience group, demonstrating advantages in calm thinking and emotion regulation [38]. Moreover, the sporting environment is inherently high-pressure and high-challenge (training, competitions, mistakes, and defeats). Athletes repeatedly experience success and failure within this "controllable high-pressure" setting, learning to view errors as opportunities for improvement. Through this process, they cognitively internalize an automatic response of "challenge = opportunity," thereby strengthening and enhancing their psychological resilience [39].

Although existing research has extensively explored EMA in college students and its influencing factors, research on how PE promotes mental health by influencing EMA remains insufficient. Particularly in the context of using EA and PR as mediating variables, these in-depth studies are crucial for understanding the role of PE in enhancing EMA among college students. As an essential manifestation of an individual's long-term participation in physical activities, EA can indirectly affect EMA by improving physical fitness and PR. In contrast, PR reflects an individual's self-regulation ability when facing stress or negative emotions and is a key mechanism for the positive impact of PE on EMA. However, how EA and PR mediate the relationship between PE and EMA in the context of PE has not been thoroughly studied. Therefore, exploring how EA and PR mediate the relationship between PE and EMA during PE is a new area worth further exploration and is expected to provide more empirical evidence and theoretical guidance for promoting the mental health of college students through PE.

This study integrates Self-Determination Theory (SDT) and Social Cognitive Theory (SCT) to propose a chain model: need satisfaction→exercise adherence (EA) → psychological resilience (PR) → emotion management ability (EMA) [40] SDT posits that satisfaction of autonomy, competence and relatedness triggers intrinsic motivation sustaining EA; SCT's agency and triadic reciprocity explain how EA, reinforced by environmental support and self-efficacy, enhances PR and subsequently EMA. PR is conceptualised dually: as a state–process capturing immediate coping strategies under stress, and as a trait–capacity denoting stable stress resistance. This bidimensional operationalisation allows measures to capture both fluid and stable facets of PR, yielding a more complete account of its mediating role in emotion regulation. In sum, the paper delineates the PE→EA/PR→EMA pathway, offering theoretical and practical guidance for university-based physical-education interventions aimed at improving students' mental health.

Based on the above, the following research hypotheses were proposed (As shown in Fig 1):

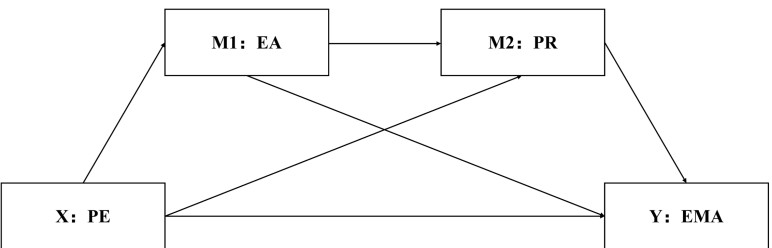

**Fig 1. Hypothesized model.** The hypothesized model illustrates a multiple mediation path with Physical Exercise (X: PE) as the independent variable and Emotional Management Ability (Y: EMA) as the dependent variable. It is proposed that physical exercise indirectly enhances emotional management ability by increasing Psychological Resilience (M1: PR) and Exercise Adherence (M2: EA), potentially achieving a moderating effect.

*H1: PE has a significant positive impact on EMA.*

*H2: EA has a significant positive mediating effect between PE and EMA.*

*H3: PR has a significant positive mediating effect between PE and EMA.*

*H4: EA and PR have a significant positive serial mediating effect between PE and EMA.*

## Methods

### Data source

The data for this study came from the Chinese College Students Physical Activity and Health Longitudinal Survey (CPAHLS-CS). CPAHLS-CS aims to collect a set of high-quality micro-data representing Chinese college students' physical activity and mental health behaviors to analyze the relevant problem areas of physical activity and health among Chinese college students and promote interdisciplinary research on college students' health issues. The sample was stratified into eight groups based on gender (male, female) and grade level (freshman, sophomore, junior, senior). The minimum sample size for each group was 45 individuals. The target sample size for each province (autonomous region, or municipality) was 1080 individuals, with an estimated national sample size (excluding Hong Kong, Macau, and Taiwan) of 33,480. The survey was uniformly distributed electronically in September 2024, via the Questionnaire Star platform, using administrative classes as units, ultimately yielding 36,756 recovered questionnaires. The final analytical sample for this study comprised students from higher education institutions in Hubei and Henan in Central China and Jiangsu and Jiangxi in East China, with 11,388 valid data points. The study protocol for this study received approval from the ethics committee at Nantong University and was documented under approval number 2022(70). Before commencing the formal investigations and testing, the researchers obtained informed consent from all the participants involved in the study. This study does not include any minors. The questionnaire in this study includes school and student number to help identify the information of the respondents. The distribution of participants is shown in Table 1.

### Measures

**Sociodemographic information.** The sociodemographic information used in this study included gender, year, etc.

**Physical exercise.** This study used the Physical Activity Rating Scale-3 (PARS-3), developed by Japanese scholar Hashimoto Kimio and revised by Liang et al. The PARS-3 examines the amount of PE among college students from three aspects: intensity, frequency, and duration of each exercise session, and uses this to measure the level of PE participation [41]. In the specific questionnaire, each item is divided into five levels, from "1" (never participate in PE) to "5" (often participate in PE). The higher the score, the greater the amount of PE. The result is a measure of PE behavior, which, to

**Table 1. Sample distribution.**

| Characteristic | | n | % |
|---|---|---|---|
| **Gender** | | | |
| | Male | 4134 | 36.3 |
| | Female | 7255 | 63.7 |
| **Grade** | | | |
| | Freshman | 7603 | 66.8 |
| | Sophomore | 2905 | 25.5 |
| | 3rd- & 4th-year Undergraduate | 880 | 7.7 |
| **Total** | | 11388 | 100 |

some extent, reflects the physical activity behavior of college students within a specific period. The raw score obtained from the questionnaire is calculated using the following formula: PE Score = Intensity × (Time – 1) × Frequency. The norm level standards for measuring PE behavior in Chinese adults are: Small amount of exercise ≤ 19 points, Moderate amount of exercise 20~42 points, Large amount of exercise > 43 points [42]. The test-retest reliability of this scale is 0.820, and its applicability in Chinese college student populations has been verified in multiple studies.

**Emotional management ability.** This study used the Emotional Intelligence Scale (EIS), translated by Chinese scholar Wang, to measure the emotional intelligence of college students [43]. The EIS examines the emotional intelligence level of college students from four dimensions: emotional perception, self-emotional management, other emotional management, and emotional application, and uses this to measure the overall state of their emotional intelligence. In the specific questionnaire, each item is divided into five levels, from "1" (completely inconsistent) to "5" (entirely consistent). The higher the score, the higher the level of emotional intelligence. The result is a quantitative measure of emotional intelligence, which, to some extent, reflects the EMA of college students. In the current academic research, its subdimensions are widely used to measure EMA [28,44,45]. This study used the EMA subscale, consisting of items 2, 6, 7, 10, 12, 14, 21, and 28 of the EIS. The Cronbach's α coefficient of the total scale was 0.907, and the structural validity was between 0.827–0.884, indicating that it has good reliability and validity and is well applicable in Chinese college student populations [46]. To further ensure the applicability of the scale among Chinese college students, we conducted a pretest in our study. By testing a portion of the sample, we confirmed that the items of the scale have a good fit with the assessment of emotional management ability in the target group and no significant cultural or group-specific biases were found.

**Exercise adherence.** This study used the PE Adherence Scale Gu of Fujian Normal University. The scale examines the PE adherence of college students from three aspects: the regularity of exercise behavior, the degree of effort invested, and the positivity of emotional experience. It uses this to measure their level of adherence to PE. In the specific questionnaire, each item is divided into five levels, from "1" (completely inconsistent) to "5" (very consistent). The higher the score, the stronger the adherence to PE. The result is a quantitative measure of PE adherence behavior, which, to some extent, reflects the PE adherence of college students within a specific period. The total scale reliability was 0.947. The fit indices were as follows: $X^2/df = 2.896 < 5.000$, CFI = 0.945 > 0.900, GFI = 0.901 > 0.800, RMSEA = 0.069 < 0.080. indicating good reliability and validity, and is suitable for assessing the PE adherence of college students [47]. In this study, we also made localization adjustments to the scale. Considering the exercise habits and cultural background of Chinese college students, we fine-tuned some items to ensure that they more closely reflect the actual experiences of the target group. The adjusted scale demonstrated good acceptability and discriminability in the pretest, further confirming its applicability among Chinese college students.

**Psychology resilience.** PR was measured using the Adolescent PR Scale developed by Hu and Gan. This scale assesses PR across five dimensions: goal focus, emotional control, positive cognition, family support, and interpersonal

assistance, providing a comprehensive measure of adolescents' PR levels. Each item in the questionnaire is rated on a 5-point Likert scale ranging from 1 (strongly disagree) to 5 (strongly agree). Higher scores indicate more excellent PR, reflecting the adolescents' adaptive capacity when facing stress and challenges. The emotional control subscale, consisting of items 1, 2, 5, 21, 23, and 27, was used for analysis in this study. Items 1, 2, 5, 21, and 27 were reverse-scored. The average score of these six items was calculated, with higher scores indicating greater emotional control within the individual's PR profile. The test-retest reliability of the scale is 0.84, demonstrating good stability and reliability, and it has been widely used in research on PR in adolescent populations [48].

## Statistical methods

Statistical analyses were performed using SPSS 26.0 and Excel. The study comprised the following steps:

(1) Data from Wen Juan Xing (a Chinese online survey platform) were pre-processed using Excel to retest or delete missing or problematic data.

(2) Harman's single-factor test assessed common method bias. Exploratory factor analysis was conducted on all electronic PE, EMA, EA, and PR variables questionnaire items. The analysis extracted six principal components with eigenvalues greater than 1, with the most prominent factor explaining 38.5% of the variance, below the commonly set threshold of 40%. Therefore, standard method bias was not considered a significant concern in this study.

(3) Descriptive statistics, including means, standard deviations, and percentages, were used to analyze participants' demographic information, including gender, grade level, ethnicity, school location, PE volume, EMA, EA, and PR. Chi-square tests were used to analyze differences in EMA among students of different genders and grade levels. Cramer's V coefficient was used to compare the strength of association between categorical variables, with values ranging from 0 to 1; higher values indicate stronger associations. A Cramer's V coefficient > 0.1 indicates a weak association, > 0.3 indicates a moderate association, and > 0.5 indicates a strong association [49]. The effect size $\eta^2$ ranges from 0 to 1, with values of 0.01 representing a small effect, 0.06 a medium effect, and 0.14 a significant impact, according to Cohen's guidelines [49].

(4) Pearson correlation analysis was used to examine the relationships among PE (low, moderate, and high volume), emotional management, EA, and PR.

(5) Regression analysis was used to test for mediation effects, with the Process macro utilized for multiple regression analysis and the bootstrap method employed for mediation effect analysis.

## Results

### Descriptive analysis

As shown in Table 2, the overall score for PE volume among college students was 17.3 ± 19.539, with significant differences observed across gender ($V = 0.412$) and grade level ($V = 0.099$) ($P < 0.001$).

As shown in Table 3, the total EMA score of undergraduates was 29.686 ± 4.413; no significant gender difference was found, whereas the difference across grades was pronounced ($P < 0.001$). Third- and fourth-year students obtained higher EMA scores (30.567 ± 4.816) than sophomores (29.257 ± 4.648). The overall EA score was 50.219 ± 9.903, with significant differences by both gender and grade ($P < 0.001$). Specifically, males reported higher EA (52.729 ± 9.804) than females (48.789 ± 8.331). By grade, third- and fourth-year undergraduates achieved the highest EA score (51.534 ± 10.423), significantly exceeding the other two grades ($P < 0.001$). The total PR score was 19.734 ± 4.572, again showing significant gender and grade effects ($P < 0.001$). Males scored higher on PR (20.614 ± 4.627) relative to females (19.233 ± 4.464). Across grades, third- and fourth-year students displayed the greatest PR (20.361 ± 5.065), significantly surpassing their younger counterparts.

**Table 2. Descriptive statistics of PE.**

| Volume Variable | | Statistical Indicator | PE Volume | | |
| --- | --- | --- | --- | --- | --- |
| | | | Small | Medium | Large |
| | | n | 8318 | 1783 | 1287 |
| | | % | 73 | 15.7 | 11.3 |
| | Male | n | 2179 | 965 | 989 |
| Gender | | % | 52.7 | 23.3 | 23.9 |
| | Female | n | 6139 | 818 | 298 |
| | | % | 84.6 | 11.3 | 4.1 |
| | | $x^2 = 1929.019$ | | | |
| | | $P < 0.001$ | | | |
| | | $V = 0.412$ | | | |
| | Freshman | n | 5569 | 1210 | 824 |
| | | % | 73.2 | 15.9 | 10.8 |
| | Sophomore | n | 2163 | 439 | 303 |
| Grade | | % | 74.5 | 15.1 | 10.4 |
| | 3rd- & 4th-year Undergraduate | n | 586 | 134 | 160 |
| | | % | 66.6 | 15.2 | 18.2 |
| | | $x^2 = 223.091$ | | | |
| | | $P < 0.001$ | | | |
| | | $V = 0.099$ | | | |

## Correlation analysis

As shown in Table 4, there was a significant positive correlation between PE and EMA ($r = 0.164$, $P < 0.001$). PE also showed significant positive correlations with EA and its sub-dimensions, with correlation coefficients ranging from 0.359 to 0.483 ($P < 0.001$). A significant positive correlation was observed between PE and PR ($r = 0.201$, $P < 0.001$). EMA was significantly positively correlated with EA and its sub-dimensions, with correlation coefficients ranging from 0.361 to 0.489 ($P < 0.001$). EMA and PR were significantly positively correlated ($r = 0.411$, $P < 0.001$). PR and EA, along with their sub-dimensions, exhibited significant positive correlations, with correlation coefficients ranging from 0.280 to 0.329 ($P < 0.001$).

## Regression analysis

Using SPSS 27.0 and PROCESS macro Model 6, a bootstrapped serial mediation analysis was conducted, controlling for gender and grade level, to examine the serial mediating effects of EA and PR between PE and EMA. This study employed methods of multicollinearity diagnostics to assess the potential for high intercorrelations among the variables, which could otherwise lead to model instability. Upon examination, the Variance Inflation Factors (VIFs) for PE, EMA, PA and PR were all found to be less than 5, indicating that there is no issue of multicollinearity in this study. The results, as shown in Table 5, indicated that after controlling for gender and grade level, PE positively predicted EA ($\beta = 0.202$, $P < 0.001$) and PR ($\beta = 0.010$, $P < 0.001$) but negatively predicted EMA ($\beta = -0.009$, $P < 0.001$). EA positively predicted PR ($\beta = 0.149$, $P < 0.001$) and EMA ($\beta = 0.206$, $P < 0.001$). PR positively predicted EMA ($\beta = 0.287$, $P < 0.001$).

   Fig 2. Path coefficients from PE to EA were 0.202 ($P < 0.001$), indicating a significant positive effect of PE on EA. Path coefficients from PE to PR were 0.010 ($P < 0.001$), indicating a significant positive effect of PE on PR. Path coefficients from PE to EMA were −0.009 ($P < 0.001$), indicating a significant negative effect of PE on EMA. Path coefficients from EA to PR were 0.149 ($P < 0.001$), indicating a significant positive effect of EA on PR. Path coefficients from EA to EMA were 0.206 ($P < 0.001$), indicating a significant positive effect of EA on EMA. Path coefficients from PR to EMA were 0.287 ($P < 0.001$), indicating a significant positive effect of PR on EMA (As show in Fig 2).

**Table 3. Descriptive statistics of EMA, EA, and PR.**

| Variable | | Statistical Indicator | EMA | EA | PR |
|---|---|---|---|---|---|
| | | M | 29.686 | 50.219 | 19.734 |
| | | SD | 4.413 | 9.093 | 4.572 |
| | | kurtosis | 1.137 | 0.949 | −0.219 |
| | | skewness | −0.153 | 0.039 | 0.191 |
| Gender | Male | M | 29.633 | 52.729 | 20.614 |
| | | SD | 4.902 | 9.804 | 4.627 |
| | | kurtosis | 0.911 | 0.708 | −0.336 |
| | | skewness | −0.227 | −0.174 | 0.150 |
| | Female | M | 29.717 | 48.789 | 19.233 |
| | | SD | 4.108 | 8.331 | 4.464 |
| | | kurtosis | 1.122 | 1.355 | −0.152 |
| | | skewness | −0.069 | 0.032 | 0.198 |
| | | η2 | <.001 | 0.043 | 0.021 |
| | | F | 0.938 | 516.842 | 245.552 |
| | | P | 0.333 | <.001 | <.001 |
| Grade | Freshman | M | 29.749 | 50.329 | 19.731 |
| | | SD | 4.251 | 8.719 | 4.553 |
| | | kurtosis | 1.195 | 0.078 | −0.243 |
| | | skewness | −0.102 | 0.965 | 0.182 |
| | Sophomore | M | 29.257 | 49.543 | 19.554 |
| | | SD | 4.648 | 9.539 | 4.445 |
| | | kurtosis | 1.146 | 1.083 | −0.075 |
| | | skewness | −0.261 | −0.047 | 0.202 |
| | 1. 3rd- & 4th-year Undergraduate | M | 30.567 | 51.534 | 20.361 |
| | | SD | 4.816 | 10.423 | 5.065 |
| | | kurtosis | 0.400 | 0.116 | −0.489 |
| | | skewness | −0.136 | 0.033 | 0.136 |
| | | η2 | 0.006 | 0.003 | 0.002 |
| | | F | 32.246 | 17.861 | 10.559 |
| | | P | <.001 | <.001 | <.001 |

**Table 4. Correlation matrix.**

| | PE | EMA | EA | Exercise Behavior | Effort Investment | Emotional Experience | PR |
|---|---|---|---|---|---|---|---|
| PE | 1 | | | | | | |
| EMA | 0.164** | 1 | | | | | |
| EA | 0.452** | 0.478** | 1 | | | | |
| Exercise Behavior | 0.483** | 0.361** | 0.882** | 1 | | | |
| Effort Investment | 0.407** | 0.460** | 0.954** | 0.776** | 1 | | |
| Emotional Experience | 0.359** | 0.489** | 0.912** | 0.665** | 0.836** | 1 | |
| PR | 0.201** | 0.411** | 0.329** | 0.313** | 0.313** | 0.280** | 1 |

**The correlation is significant at the 0.01 level (two-tailed).

**Table 5. Regression analysis results with covariates included.**

| Regression | | Fitting indices | | | Coefficient | | |
| --- | --- | --- | --- | --- | --- | --- | --- |
| Outcome variables | Predictive variables | R | R² | F | β | SE | t |
| EMA | | 0.566 | 0.32 | 1071.485*** | | | |
| | PE | | | | −0.009 | 0.002 | −4.536*** |
| | EA | | | | 0.206 | 0.004 | 47.203*** |
| | PR | | | | 0.287 | 0.008 | 36.177*** |
| | Gender | | | | 1.152 | 0.077 | 14.987*** |
| | Grade | | | | 0.006 | 0.051 | 0.107 |
| EA | | 0.454 | 0.206 | 986.794*** | | | |
| | PE | | | | 0.202 | 0.004 | 48.306*** |
| | Gender | | | | −0.891 | 0.171 | −5.226*** |
| | Grade | | | | −0.078 | 0.113 | −0.688 |
| PR | | 0.341 | 0.116 | 374.466*** | | | |
| | PE | | | | 0.01 | 0.002 | 3.964*** |
| | EA | | | | 0.149 | 0.005 | 29.936*** |
| | Gender | | | | −0.661 | 0.906 | −7.299*** |
| | Grade | | | | 0.134 | 0.06 | 2.221 |

Note: ***$P<0.001$.

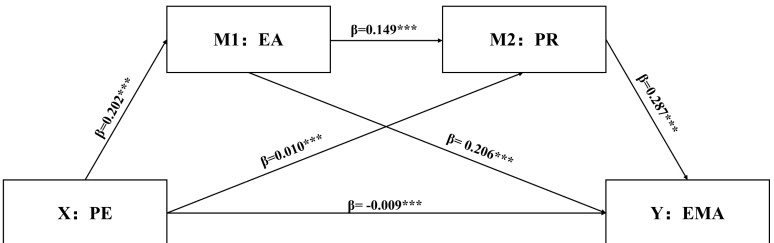

**Fig 2. Mediation model.** This mediation model uses physical exercise (PE) as the independent variable and emotional management ability (EMA) as the dependent variable, with exercise adherence (EA) and psychological resilience (PR) as parallel mediators. Path coefficients show that PE→EA (β=0.149, $P<0.001$), EA→EMA (β=0.206, $P<0.001$), PR→EMA (β=0.287, $P<0.001$), and PE→PR (β=−0.009, $P<0.001$) are all significant. The results indicate that EA mediates the relationship between PE and EMA. Note:***$P<0.001$.

The correlation analysis between PE and EMA shows a significant positive correlation (r=0.164, $P<0.01$), which seems to indicate that PE has a positive influence on EMA. However, the regression analysis reveals a completely different situation, with PE having a significant negative direct effect on EMA (β=−0.009, $P<0.001$). Therefore, we conducted a stepwise regression analysis as shown in Table 6. The results show that the root of this contradiction may lie in the mediating variables EA and PR. They carry a large amount of effect in this system. In the path from PE to EMA, EA and PR not

**Table 6. Stepwise regression model.**

| EMA | | Model 1 | Model 2 | | | Model 3 | | | | |
| --- | --- | --- | --- | --- | --- | --- | --- | --- | --- | --- |
| | | PE | PE | PR | EA | PE | PR | EA | Gender | Grade |
| | B | 0.037 | −0.019 | 0.204 | 0.279 | −0.009 | 0.206 | 0.287 | 1.152 | 0.005 |
| | t | 17.773 | −9.633 | 46.326 | 34.918 | −4.536 | 47.203 | 36.177 | 14.987 | 0.107 |
| | VIF | 1 | 1.262 | 1.358 | 1.126 | 1.402 | 1.359 | 1.132 | 1.174 | 1.008 |

only play mediating roles individually but also form a chain mediating path. When these mediating variables are taken into account, their mediating effects are substantial, masking the true direct effect of PE on EMA, thus presenting a positive correlation in the overall correlation analysis. However, in the regression analysis, after controlling for these mediating variables, the direct effect of PE on EMA emerges, resulting in a negative direct effect.

## Mediation effect analysis

As shown in Table 7, the 95% confidence interval (CI) for the mediation effect of EA between PE and EMA was [0.039, 0.045], excluding zero. Therefore, in this study, EA significantly mediated the relationship between PE and EMA. The 95% CI for the mediation effect of PR between PE and EMA was [0.001, 0.004], excluding zero, indicating that PR significantly mediated the relationship between PE and EMA. Furthermore, the 95% CI for the serial mediation effect of EA and PR between PE and EMA was [0.008, 0.010], indicating a significant serial mediation effect.

## Discussion

This study aimed to elucidate the relationship between PE and EMA among college students. By incorporating the variables of EA and PR, a serial mediation model was constructed and validated to examine the relationship between PE and EMA. The findings revealed a significant serial mediation effect of EA and PR between PE and EMA in college students. By constructing and empirically validating a serial mediation model linking "physical exercise -- exercise adherence -- psychological resilience -- emotional management ability," this study is the first, within a Chinese undergraduate population, to integrate core constructs from the disciplines of physical education, psychology, and education into a single, evidence-based framework. The findings imply that, if universities adopt this framework to systematically restructure their existing physical-education and mental-health systems, they can not only achieve a marked reduction in population-level incidence of emotional disorders such as depression and anxiety, but also cultivate a new archetype of graduate who possesses lifelong exercise habits, elevated psychological resilience, and superior emotional-regulation skills. Consequently, the educational value of physical education is elevated beyond mere physical enhancement toward personality development and character formation, thereby fully realizing its contemporary mandate of "educating through sport, strengthening the mind through sport, and forging character through sport."

## Descriptive results

The descriptive statistics for all variables in this study revealed significant differences in EMA among college students across different academic years. However, no significant gender differences were observed, a finding inconsistent with prior research indicating gender-based variations in EMA among college students [10]. Previous studies have reported significant gender differences in EMA during early and mid-adolescence [50]. These disparities may stem from genetic, biological, hormonal, cultural, and social influences [51]. It is plausible that gender differences in EMA diminish with increasing age and cognitive maturation [52]. Furthermore, significant differences in PE, EA, and PR were observed across gender and academic years, aligning with previous research findings [53–56].

**Table 7. Presents the mediation effect analysis.**

|  | Effect size | *BootSE* | 95% CI |
| --- | --- | --- | --- |
| Total Effect | 0.044 | 0.002 | [0.039,0.0482] |
| Direct Effect | −0.009 | 0.002 | [-0.013,-0.050] |
| Indirect Effect | 0.053 | 0.002 | [0.05,0.057] |
| PE→EA→EMA | 0.042 | 0.002 | [0.039,0.045] |
| PE→PR→EMA | 0.003 | 0.001 | [0.001,0.004] |
| PE→EA→PR→EMA | 0.009 | 0.001 | [0.008,0.010] |

When examining the relationship between PE volume and EMA, the study revealed a significant negative predictive relationship (β = −0.009), failing to support research hypothesis H1. However, correlation analysis indicated a significant positive correlation between these variables (r = 0.164). This seemingly contradictory finding may arise from the combined effects of multiple factors. The negative prediction coefficient in the regression analysis may suggest the presence of confounding variables. Within the multiple regression model, PE volume may exhibit collinearity with other variables (e.g., stress levels and sleep quality). These variables might be prioritized in the model, thereby attenuating the direct influence of PE volume on EMA. Conversely, the positive correlation in the correlation analysis may reflect a direct association between PE volume and EMA, suggesting that PE can enhance EMA. When controlling for additional variables, other, more prominent factors might obscure or reverse this direct relationship. Sample characteristics, the sensitivity of measurement tools, and limitations inherent in the statistical methods employed may also contribute to this inconsistency. Therefore, when interpreting this phenomenon, carefully considering model complexity, variable interactions, and potential biases within the study design is warranted to elucidate the complex relationship between PE volume and EMA fully. In the context of sports development, enhancing physical fitness and regulating emotions through PE is particularly crucial.

## Positive mediating effect of exercise adherence

This study found that EA mediates the relationship between PE and EMA in college students, with a significant positive mediating effect size of 0.042, supporting Hypothesis H2. Furthermore, EA and its sub-dimensions positively correlated significantly with PE and EMA. This indicates that EA is a crucial link between PE and EMA in college students. EA, a key psychological variable, is frequently employed in various research studies [57]. In this study, EA was examined as a mediating variable to investigate the mechanism through which PE influences EMA. The results revealed that the positive mediating effect of EA enhances the positive effects of PE on EMA. This finding is consistent with previous research, which suggests that regular exercise can alleviate anxiety and stress by regulating stress hormones (such as cortisol), enhancing emotional stability and self-regulation, and thereby positively influencing individual development and adaptation [58,59].

College students who regularly engage in PE can better maintain their EA through positive physical experiences and enhanced self-efficacy during workouts [60]. This EA improves physical fitness and promotes emotional regulation and self-efficacy through physical activity, ultimately improving EMA [61]. While maintaining EA, individuals gradually develop a positive psychological state that not only aids their progress in exercise but also facilitates the transfer of this positive self-regulation ability to other areas of life [62,63]. EA provides individuals with a stable self-regulation mechanism, enabling them to adjust their mindset more effectively when facing emotional fluctuations [64]. EA plays a crucial mediating role between PE and EMA through the combined action of multiple factors. This mediating effect is not only effective in the short term. Still, it is also a continuous and positive process, helping individuals form healthy self-regulation patterns and long-term psychological adaptation abilities [65]. Therefore, interventions that strengthen EA are essential for promoting overall psychological well-being and emotional regulation ability in individuals. Promoting and maintaining individuals' EA behaviors makes it possible to improve their PE outcomes and comprehensively enhance their EMA and overall psychological health. This is of great significance for cultivating healthy individuals who can adapt to the pressures of modern society.

## Positive mediating effect of psychological resilience

This study revealed a positive mediating effect of PR between PE and EMA in college students (effect size = 0.003, **P** < 0.05), supporting Hypothesis H3. Furthermore, significant positive correlations were found between PR, PE and EMA. The results indicate that the enhancement of PR among college students in PE has a positive impact on their EMA. This aligns with previous research [66], demonstrating a significant positive correlation between PR and EMA. Specifically, college students with higher levels of PR are more likely to exhibit greater confidence in their ability to complete tasks and achieve desired goals. They demonstrate an enhanced capacity to perceive and evaluate emotions in specific situations.

Individuals who regularly participate in PE experience a significant enhancement in their PR through the physical and psychological experiences gained during workouts [67,68]. This enhanced PR further facilitates their EMA, enabling them to more effectively regulate and manage their emotional states when faced with stress and emotional fluctuations [69]. As a positive mind-body activity, PE promotes the release of endorphins and other neurotransmitters, helping to alleviate stress and anxiety and improve emotional stability and psychological well-being. In this process, PR plays a crucial mediating role [70]. During exercise, individuals gradually learn to control their emotional responses through physical activity and mental relaxation. This PR helps them maintain a positive mindset during exercise and transfer this ability to their daily lives, enabling them to manage their emotions more effectively when facing stress and challenges [66]. Research indicates a significant association between emotional regulation self-efficacy and EA; increasing PR can enhance an individual's EA, improving EMA [71]. The array of positive effects resulting from PR extends beyond improvements in physical fitness and emotional management, representing a holistic approach to enhancing overall quality of life.

### Serial mediation of exercise adherence and psychological resilience

This study revealed a serial mediating effect of EA and PR between PE and EMA (effect size = 0.009, *P* < 0.05), supporting Hypothesis H4. Correlation analysis indicated that EA and its sub-dimensions, as well as PR, exhibited significant positive correlations with both PE and EMA, further validating the accuracy of the serial mediation model. These results suggest that PE enhances EA, which, in turn, strengthens PR, ultimately leading to a positive promotion of EMA among college students.

PE positively influences EMA by enhancing an individual's EA. According to Bandura's social cognitive theory, an individual's self-efficacy significantly predicts their behavior. PE can improve individuals' belief in and behavior of adhering to exercise by increasing self-efficacy [72]. This adherence not only contributes to the improvement of physical fitness but also improves emotional states through both physiological and psychological mechanisms. Enhanced EA further indirectly influences EMA through the mediating variable of PR. Research indicates that regular PE can significantly improve an individual's PR, helping them better manage negative emotions such as anxiety and depression [73]. Furthermore, according to Abraham Maslow's hierarchy of needs theory, PE satisfies an individual's essential health and safety needs and supports higher-level needs, such as self-actualization [74]. Through the serial mediating effect of EA and PR, PE can help individuals better realize their self-worth, enhancing overall EMA [75]. Victor Vroom's expectancy theory also supports the improvement of EMA through PE. According to this theory, an individual's expectation of behavioral outcomes and their evaluation of the value of those outcomes jointly influence their behavioral motivation [76]. The positive emotional experiences and health benefits PE brings can enhance individuals' expectations of improving their EMA, promoting adherence to exercise behavior. Therefore, PE, through the serial mediation path of EA and PR, ultimately promotes more effective EMA, providing a meaningful way to improve college students' overall health and well-being.

Limitations of this study: (1) Testing using recall-based scales without instrumental testing. (2) The analysis framework may have significant endogeneity issues, including potential reverse causality. This complicates the direction of causal relationships between explanatory variables, which may lead to misunderstandings of the analysis. Even if statistical correlations exist, they should be interpreted cautiously, as the exact causal relationships remain unclear. Future research may address these limitations by employing longitudinal designs or experimental approaches (e.g., randomized controlled trials), including further exploration of potential variables to better establish causal relationships.

### Conclusion

This study demonstrates that physical exercise (PE) enhances emotional management ability (EMA) not directly, but through a serial mediation chain. Specifically, regular physical activity first strengthens students' exercise adherence (EA), which reflects behavioral self-discipline and commitment. The successful experiences and physiological improvements resulting from sustained exercise (such as increased physical fitness and reduced stress) subsequently enhance their

psychological resilience (PR)—the capacity to adapt and rebound in the face of pressure and adversity. Ultimately, this fortified psychological resilience enables students to employ more positive cognitive reappraisal strategies and improved impulse control to manage daily emotions (EMA). This finding suggests universities should incorporate PE as an integral component of mental health education to facilitate students' self-actualization and improve their overall psychological well-being.

## Acknowledgments

We sincerely thank all the staff and students from the participating schools and our co-operators for their assistance in data collection.

## Author contributions

**Data curation:** Yu-han Li, Wei-dong Zhu, Hu Lou, Bo Li.

**Formal analysis:** Wei-dong Zhu, Yu-yan Qian.

**Investigation:** Yu-han Li, Wei-dong Zhu, Hu Lou, Yu-yan Qian.

**Project administration:** Yu-han Li, Hu Lou, Bo Li, Yu-yan Qian.

**Resources:** Yu-han Li, Wei-dong Zhu, Hu Lou, Bo Li, Yu-yan Qian.

**Visualization:** Wei-dong Zhu.

**Writing – original draft:** Shan-shan Han, Yu-han Li.

**Writing – review & editing:** Shan-shan Han, Yu-han Li.

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
