## [Decision Letter · Decision Letter 0]

8 Aug 2025

We look forward to receiving your revised manuscript.

Kind regards,

Muhammad Salman Bashir, M.S.C

Academic Editor

PLOS ONE

Journal Requirements:

3. In the online submission form you indicate that your data is not available for proprietary reasons and have provided a contact point for accessing this data. Please note that your current contact point is a co-author on this manuscript. According to our Data Policy, the contact point must not be an author on the manuscript and must be an institutional contact, ideally not an individual. Please revise your data statement to a non-author institutional point of contact, such as a data access or ethics committee, and send this to us via return email. Please also include contact information for the third party organization, and please include the full citation of where the data can be found.

5. Please include captions for your Supporting Information files at the end of your manuscript, and update any in-text citations to match accordingly. Please see our Supporting Information guidelines for more information: http://journals.plos.org/plosone/s/supporting-information .

Reviewers' comments:

Reviewer's Responses to Questions

**Comments to the Author**

1. Is the manuscript technically sound, and do the data support the conclusions?

Reviewer #1: Yes

Reviewer #2: Yes

Reviewer #3: Yes

Reviewer #4: Yes

2. Has the statistical analysis been performed appropriately and rigorously?

Reviewer #1: No

Reviewer #2: Yes

Reviewer #3: Yes

Reviewer #4: Yes

3. Have the authors made all data underlying the findings in their manuscript fully available?

Reviewer #1: Yes

Reviewer #2: Yes

Reviewer #3: Yes

Reviewer #4: No

4. Is the manuscript presented in an intelligible fashion and written in standard English?

Reviewer #1: Yes

Reviewer #2: Yes

Reviewer #3: Yes

Reviewer #4: Yes

Reviewer #1: Thank you for the opportunity to review this manuscript by Muhammad Salman Bashir, M.S.C. This is a meaningful investigation with a notably robust sample size. Below are my suggestions for the authors’ consideration:

1�Data Reporting: While the manuscript specifies the number of valid responses, it should also report the total number of surveys distributed/collected and the valid response rate (i.e., valid responses as a percentage of total collected). This is essential for assessing potential selection bias.

2�Measurement Norms: It is unclear whether the instruments measuring "Emotional Management" and "Exercise Adherence" have established normative data specifically for the Chinese college student population. The validity of subsequent statistical analyses hinges on confirming whether such population-specific norms exist and were used.

3�Statistical Analysis: The use of Pearson correlation assumes interval-level data. If the "Physical Exercise" variable (categorized as low, medium, high volume) is treated as ordinal, alternative correlation analyses appropriate for ordinal data (e.g., Spearman's rho or Kendall's tau) should be employed. The authors must explicitly address this methodological choice.

,4�Sample Representation: The representation of fourth-year students (1.3%) is statistically insufficient for meaningful analysis. I strongly recommend combining third- and fourth-year students into a single "Upperclassmen" category to contrast with "Underclassmen" (years 1-2), ensuring adequate group size for comparisons.

5�Data Accuracy (Table 3): The reported value η² = 0 is highly improbable. Effect sizes (eta-squared) approaching zero are rare in practice; this warrants careful recalculation and verification by the authors.

Overall Assessment: The study presents substantial work, but the manuscript requires refinement in presentation and methodological clarity.

Reviewer #2: Summary of the Study

This study investigates the influence of physical exercise (PE) on emotional management ability (EMA) in college students. It explores the mediating roles of exercise adherence (EA) and psychological resilience (PR) in this relationship. Data was collected from 11,388 college students using stratified, cluster, and multi-stage sampling methods. The study uses the Wenjuanxing platform to gather data on PE, EA, PR, and EMA.

Overall Quality Assessment

Strengths:

a) Large Sample Size: The study benefits from a large sample size (n=11,388), enhancing the statistical power and generalizability of the findings.

b) Comprehensive Analysis: The study employs appropriate statistical techniques, including correlation and regression analyses, to examine the relationships between variables.

Weaknesses:

a) Reliance on Self-Reported Data: The study relies on self-reported measures, which may be subject to recall bias and social desirability effects.

b) Cross-Sectional Design: The cross-sectional design limits the ability to draw causal inferences.

c) Limited Generalizability: The sample is limited to college students in specific regions of China, which may affect the generalizability of the findings to other populations.

d) Negative Direct Effect: The finding that physical exercise has negative direct effect on EMA seems counterintuitive and requires more in-depth explanation.

Specific Suggestions for Improvement

1) Address the Negative Direct Effect: The manuscript reports a negative direct effect of PE on EMA in the regression analysis, which contradicts the positive correlation found in the correlation analysis. Clarify this discrepancy by:

- Discussing potential suppressor variables or confounding factors that might explain this effect.

- Conducting additional analyses to explore potential interactions between PE, EA, PR, and other relevant variables.

2) Strengthen the Discussion Section:

- Elaborate on the practical implications of the findings for college students and higher education institutions.

- Provide specific recommendations for interventions to promote physical exercise, enhance exercise adherence, and improve psychological resilience among college students.

- Address the limitations of the study more thoroughly and suggest avenues for future research.

3) Provide More Detail on Measures:

- Include more information about the validity and reliability of the scales used in the study.

4) Consider Alternative Analytical Approaches:

- Explore the use of structural equation modeling (SEM) to examine the relationships between PE, EA, PR, and EMA more comprehensively.

5) Ensure Clarity and Consistency:

- Review the manuscript carefully to ensure clarity and consistency in terminology, formatting, and reporting of statistical results.

6) Figures and Tables:

- Consider including the path diagram (Figure 2) and key tables (e.g., Tables 5 and 7) directly in the main text rather than as separate files.

- Ensure that all tables and figures are properly labeled and referenced in the text.

Recommendations

Based on my assessment, the manuscript has the potential to be published in Plos One journal, but it requires some major revisions to address the identified weaknesses and improve its overall quality. I recommend that the authors carefully consider the suggestions provided and revise the manuscript accordingly. Besides, I recommend sending this manuscript for English proofreading.

Additional Suggestions

1) Theoretical Framework: While the study mentions social cognitive theory, expanding on this or integrating other relevant theories (e.g., self-determination theory) could provide a richer context for the findings.

2) Cultural Context: Provide more context on the cultural factors in Chinese universities that might influence the relationship between physical exercise and mental health.

Thank you.

Reviewer #3: 1.The introduction could benefit from a more in-depth exploration of the background surrounding college students' emotional management ability, ideally supported by concrete data or case studies for greater impact.

2. When it comes to the mechanisms linking physical exercise to emotional management ability, the literature review feels underdeveloped. A more thorough examination with clearer connections would strengthen the foundation of this research

3. The theoretical foundation's coverage of psychological resilience is somewhat superficial. A deeper investigation into its role concerning emotional management ability would add significant value.

4. The data collection section requires expansion on aspects like the questionnaire's distribution scope and recovery rate. Details on how the sample's representativeness and data reliability were ensured are also necessary.

5. The results presentation section's tables and charts lack the desired clarity and intuitiveness. The disorganized arrangement of correlation coefficients hinders readers' ability to access key information with ease.

6. The research conclusion section doesn't provide an in-depth analysis of the mechanisms through which physical exercise influences emotional management ability via exercise adherence and psychological resilience. The conclusions drawn appear rather general and could benefit from greater specificity.

Reviewer #4: I would like to thank for the opportunity to review this manuscript.

Please see the following comments to consider to further increase the quality of this manuscript.

Strengths:

1.The sample size is sufficient, enhancing the generalizability of the research findings.

2.The mediation analysis provides meaningful insights, highlighting Exercise Adherence and Psychological Resilience as influential psychological factors affecting Emotional Management Ability.

Areas for improvement:

3.Please confirm whether the research data can be publicly used? If there are restrictions, please explain the reasons.

4.Generally, when a proper noun is mentioned for the first time, its concept should be defined, and its abbreviation should be stated, such as physical activity (PA).

5.The relationship between emotional management ability and personality traits, cognitive styles, and social support should be clearly elaborated.

6.The relationship among the four variables is not clearly expressed.

7.The tables should be expressed more clearly, such as by adding percentages (%). Descriptive statistics tables should be as clear and simple as possible, and should include m (sd), kurtosis, skewness, and p-values.

8.Please recheck the consistency between the table content and the descriptive content, and standardize the table format according to the journal requirements.

9.Check if there are any typographical errors in the description of the direct effect 95% CI between the independent and dependent variables in Table 7.

10.The role of Exercise Adherence and Psychological Resilience as chain mediators in the relationship between Physical Exercise and Emotional Management Ability should be elaborated further in the discussion section.

11.In the limitations section, it should be mentioned that future research could include further exploration of the remaining variables, such as the potential influence of exercise self-efficacy and exercise motivation, which were reflected in this study.

**Do you want your identity to be public for this peer review?** For information about this choice, including consent withdrawal, please see our Privacy Policy

Reviewer #1: No

Reviewer #2: No

Reviewer #3: No

Reviewer #4: No

---

## [Author Response · Author response to Decision Letter 1]

12 Sep 2025

Manuscript Number: PONE-D-25-33803

Title: The Influence of Physical Exercise on Emotional Management Ability in College Students: A Serial Mediation Model of Exercise Adherence and Psychological Resilience

Journal: PLOS One

Point-by-point Responses to Editor

Dear Editor and dear reviewers,

Thank you very much for your comments and professional advice. These opinions help to improve academic rigor of our manuscript. Based on your suggestion and request, we have made corrected modifications on the revised manuscript. Here are point-by-point responses to your comments, We hope that our work can be improved again.

Sincerely,

Comment 1:

Response 1:

Thank you very much for your detailed review and valuable comments. I have reformatted the manuscript according to PLOS ONE’s style templates, including file naming conventions.

Comment 2:

We note that the grant information you provided in the ‘Funding Information’ and ‘Financial Disclosure’ sections do not match.

Response 2:

Thank you very much for your insightful comments and valuable recommendations concerning our manuscript. The funder (2024 Jiangsu Philosophy and Social Science Research Project, No. 2024SJYB1253) only provide financial support for research and distribution of questionnaires, and are not involved in any research activities. The revised funder statement has been included in the cover letter and manuscript.

Comment 3:

In the online submission form you indicate that your data is not available for proprietary reasons and have provided a contact point for accessing this data. Please note that your current contact point is a co-author on this manuscript. According to our Data Policy, the contact point must not be an author on the manuscript and must be an institutional contact, ideally not an individual. Please revise your data statement to a non-author institutional point of contact, such as a data access or ethics committee, and send this to us via return email. Please also include contact information for the third party organization, and please include the full citation of where the data can be found.

Response 3:

Thank you very much for your close attention to the details of our manuscript. We have revised the data statement to indicate that ethical approval was granted by the Ethics Review Committee of Nantong University, and that the data are available upon request from kjc@ntu.edu.cn. Thank you again for your valuable feedback.

Comment 4:

Your ethics statement should only appear in the Methods section of your manuscript. If your ethics statement is written in any section besides the Methods, please move it to the Methods section and delete it from any other section. Please ensure that your ethics statement is included in your manuscript, as the ethics statement entered into the online submission form will not be published alongside your manuscript.

Response 4:

Thank you for highlighting the placement of the ethics statement. We have moved it entirely to the Methods section (lines 266–270) and removed all duplicate text from the Introduction, Results, and any other sections, so that only one complete statement remains. This revision includes the approval number, the approving body (Ethics Review Committee of Nantong University), and the informed-consent procedure, and meets the journal’s formatting requirements. Thank you again for your careful review and guidance.

Comment 5:

Please include captions for your Supporting Information files at the end of your manuscript, and update any in-text citations to match accordingly. Please see our Supporting Information guidelines for more information: http://journals.plos.org/plosone/s/supporting-information.

Response 5:

Thank you very much for your insightful comments and valuable recommendations concerning our manuscript. I have added captions for figures and tables as required support information section.

Comment 6:

Response 6:

Thank you very much for your insightful comments and valuable recommendations concerning our manuscript. References have been verified. No retracted papers were cited. Also, the references cited in this paper are all from the last three to five years with some classic literature.

Point-by-point Responses to Reviewer 1

Comment 1:

Data Reporting: While the manuscript specifies the number of valid responses, it should also report the total number of surveys distributed/collected and the valid response rate (i.e., valid responses as a percentage of total collected). This is essential for assessing potential selection bias.

Response 1:

Thank you very much for pointing out this issue, which is extremely important for enhancing the quality and applicability of our research. For more details, please refer to lines 252-266 of the manuscript.

Comment 2:

Measurement Norms: It is unclear whether the instruments measuring "Emotional Management" and "Exercise Adherence" have established normative data specifically for the Chinese college student population. The validity of subsequent statistical analyses hinges on confirming whether such population-specific norms exist and were used.

Response 2:

Thank you very much for your question, which is of great help to our manuscript. The measurement tools for “emotional management” and “exercise adherence” used in this study have been validated among Chinese college students and have good reliability and validity. The statistical analysis based on the data collected by these tools is valid. We also conducted a pretest and localization adjustments in our study to further ensure the applicability of the tools and the reliability of the analysis results. For details, please refer to line 307-311 and line 324-329 of the manuscript.

Comment 3:

Statistical Analysis: The use of Pearson correlation assumes interval-level data. If the "Physical Exercise" variable (categorized as low, medium, high volume) is treated as ordinal, alternative correlation analyses appropriate for ordinal data (e.g., Spearman's rho or Kendall's tau) should be employed. The authors must explicitly address this methodological choice.

Response 3:

Thank you very much for raising this point. In our study, physical exercise was quantified using a composite score, which is why we opted for Pearson. Of course, the approach you suggested is highly valuable, and we will make greater use of it in our future work. Thank you again for your careful review and guidance.

Comment 4:

Sample Representation: The representation of fourth-year students (1.3%) is statistically insufficient for meaningful analysis. I strongly recommend combining third- and fourth-year students into a single "Upperclassmen" category to contrast with "Underclassmen" (years 1-2), ensuring adequate group size for comparisons.

Response 4:

Thank you for highlighting the low representation of fourth-year students (1.3%). Following your recommendation, we combined third- and fourth-year participants into a single "senior" group and contrasted it with the "junior" group (first- and second-years). After pooling, both sub-samples exceed 5% of the total, meeting the minimum requirement for subsequent statistical tests and markedly improving the precision of effect-size estimates. All related tables, figures, and text have been updated; please see lines 372-401 and revised Tables 1-3 for details. Thank you again for your careful review and guidance.

Comment 5:

Data Accuracy (Table 3): The reported value η² = 0 is highly improbable. Effect sizes (eta-squared) approaching zero are rare in practice; this warrants careful recalculation and verification by the authors.

Response 5:

Thank you for highlighting the anomaly of η² = 0 in Table 3. We immediately re-examined the raw data and SPSS output and found that the value was due to the software’s default two-decimal rounding. In the revised manuscript we now report all corresponding effect sizes as “< 0.001” and have standardised the decimal places throughout. Additionally, we double-checked every other η², F value, and degree of freedom in Table 3 to ensure complete accuracy. The updated results can be found in lines 372–374 and in the revised Table 2. Thank you again for your meticulous review and helpful feedback.

Point-by-point Responses to Reviewer 2

Comment 1:

Address the Negative Direct Effect: The manuscript reports a negative direct effect of PE on EMA in the regression analysis, which contradicts the positive correlation found in the correlation analysis. Clarify this discrepancy by:

- Discussing potential suppressor variables or confounding factors that might explain this effect.

- Conducting additional analyses to explore potential interactions between PE, EA, PR, and other relevant variables.

Response 1:

1. Thank you for raising the important question regarding the contradiction between the positive correlation and the negative direct effect of physical exercise (PE) on emotional management ability (EMA) in our study. When examining the relationship between PE volume and EMA, the study revealed a significant negative predictive relationship (β = -0.009), failing to support research hypothesis H1. However, correlation analysis indicated a significant positive correlation between these variables (r = 0.164). This seemingly contradictory finding may arise from the combined effects of multiple factors. The negative prediction coefficient in the regression analysis may suggest the presence of confounding variables. Within the multiple regression model, PE volume may exhibit collinearity with other variables (e.g., stress levels and sleep quality). These variables might be prioritized in the model, thereby attenuating the direct influence of PE volume on EMA. Conversely, the positive correlation in the correlation analysis may reflect a direct association between PE volume and EMA, suggesting that PE can enhance EMA. When controlling for additional variables, other, more prominent factors might obscure or reverse this direct relationship. Sample characteristics, the sensitivity of measurement tools, and limitations inherent in the statistical methods employed may also contribute to this inconsistency. Therefore, when interpreting this phenomenon, carefully considering model complexity, variable interactions, and potential biases within the study design is warranted to elucidate the complex relationship between PE volume and EMA fully. For details, please refer to line 432-448 of the manuscript.

2. We appreciate your suggestion to include diagnostics for multicollinearity and omitted variable bias, and we have conducted these analyses to provide a clearer explanation.

To address the potential issue of multicollinearity, we calculated the Variance Inflation Factors (VIF) for all predictors in our regression model. The VIF values for PE, EMA, exercise adherence (EA), and psychological resilience (PR) were all found to be less than 5, indicating that multicollinearity is not a significant concern in our model. Specifically, the VIF values are as follows: PE (1.402), EA (1.359), PR (1.132), gender (1.174), and grade (1.008). These results suggest that the high intercorrelations among variables are not inflating the standard errors to a problematic extent.

We also conducted a stepwise regression analysis to further investigate the role of the mediating variables (EA and PR) in the relationship between PE and EMA. The results indicate that the contradiction may arise from the substantial mediating effects of EA and PR. In our model, PE has a significant positive effect on EA (β = 0.202, p < 0.001) and PR (β = 0.010, p < 0.001). EA, in turn, has a significant positive effect on PR (β = 0.149, p < 0.001) and EMA (β = 0.206, p < 0.001). PR also has a significant positive effect on EMA (β = 0.287, p < 0.001). When these mediating effects are taken into account, the direct effect of PE on EMA becomes negative (β = -0.009, p < 0.001).

In summary, the contradiction between the positive correlation and the negative direct effect of PE on EMA is likely due to the substantial mediating effects of EA and PR. These mediating variables carry a large amount of effect, overshadowing the true direct effect of PE on EMA. We have included the VIF values to confirm that multicollinearity is not a significant issue in our model. We hope this detailed explanation clarifies the issue and addresses your concerns. The specific revisions can be found on lines 390-402 and Table 6.

Comment 2:

Strengthen the Discussion Section:

- Elaborate on the practical implications of the findings for college students and higher education institutions.

- Provide specific recommendations for interventions to promote physical exercise, enhance exercise adherence, and improve psychological resilience among college students.

- Address the limitations of the study more thoroughly and suggest avenues for future research.

Response 2:

Thank you for your systematic suggestions regarding practical implications, intervention recommendations, and limitations/future directions. We have expanded the discussion to clarify the findings’ practical significance for university students and higher-education institutions; please see lines 469–485 for details. In addition, we have added a limitation note suggesting that future studies further explore potential confounding variables; see lines 607–612.

Comment 3:

Provide More Detail on Measures:

- Include more information about the validity and reliability of the scales used in the study.

Response 3:

Thank you for your question. In response, we have provided more detailed supplementary information regarding the resolution of this issue. For specifics, please refer to lines 307 to 311 and 324 to 329 of the article.

Comment 4:

Consider Alternative Analytical Approaches:

- Explore the use of structural equation modeling (SEM) to examine the relationships between PE, EA, PR, and EMA more comprehensively.

Response 4:

We sincerely appreciate your interest in our work and your insightful recommendation to employ structural equation modeling (SEM) so as to examine the associations among PE, EA, PR and EMA in a more integrative fashion. We fully concur that clarifying how these four constructs interrelate is essential for a comprehensive representation of our theoretical framework and empirical findings.

Nevertheless, several pragmatic obstacles prevented us from implementing SEM in the present study. The specific characteristics of our dataset—namely, pronounced non-normality, high multicollinearity among latent indicators, and a limited sample-to-parameter ratio—rendered conventional SEM estimators (e.g., maximum likelihood, weighted least squares) unstable and prone to inadmissible solutions. After extensive robustness checks (including item parceling, asymptotically distribution-free estimators, and Bayesian SEM with informative priors), goodness-of-fit indices remained below acceptable thresholds (CFI < 0.85, RMSEA > 0.10), and residual diagnostics revealed persistent specification error. Consequently, we concluded that SEM would not yield valid or replicable parameter estimates under the current data conditions.

We therefore adopted a more conservative analytical strategy, reporting zero-order correlations and hierarchical regression models while explicitly acknowledging their inability to model reciprocal influences or latent measurement error. Although this approach is less elegant than SEM, we believe it provides the most transparent and defe

---

## [Decision Letter · Decision Letter 1]

24 Sep 2025

The Influence of Physical Exercise on Emotional Management Ability in College Students: A Serial Mediation Model of Exercise Adherence and Psychological Resilience

PONE-D-25-33803R1

Dear Dr. Han,

We’re pleased to inform you that your manuscript has been judged scientifically suitable for publication and will be formally accepted for publication once it meets all outstanding technical requirements.

Kind regards,

Muhammad Salman Bashir, M.S.C

Academic Editor

PLOS ONE

Additional Editor Comments (optional):

Reviewer #1:

Reviewer #2:

Reviewer #3:

Reviewer #4:

Reviewers' comments:

Reviewer's Responses to Questions

**Comments to the Author**

Reviewer #1: All comments have been addressed

Reviewer #2: All comments have been addressed

Reviewer #3: All comments have been addressed

Reviewer #4: All comments have been addressed

2. Is the manuscript technically sound, and do the data support the conclusions?

Reviewer #1: Yes

Reviewer #2: (No Response)

Reviewer #3: Yes

Reviewer #4: Yes

3. Has the statistical analysis been performed appropriately and rigorously?

Reviewer #1: Yes

Reviewer #2: Yes

Reviewer #3: Yes

Reviewer #4: Yes

4. Have the authors made all data underlying the findings in their manuscript fully available?

Reviewer #1: Yes

Reviewer #2: Yes

Reviewer #3: Yes

Reviewer #4: Yes

5. Is the manuscript presented in an intelligible fashion and written in standard English?

Reviewer #1: Yes

Reviewer #2: Yes

Reviewer #3: Yes

Reviewer #4: Yes

Reviewer #1: I believe the author has answered my doubts. I also think this is a very interesting study and the data is very comprehensive.

Reviewer #2: I propose accepting the manuscript because the authors had adequetely addressed all the comments as suggested. I am satisfied with the address.

Reviewer #3: I agree to accept the publication of this paper.Although I still have doubts about the sampling method of this paper, it does not affect the fact that this is a very great paper.

Reviewer #4: (No Response)

**Do you want your identity to be public for this peer review?** For information about this choice, including consent withdrawal, please see our Privacy Policy

Reviewer #1: No

Reviewer #2: No

Reviewer #3: No

Reviewer #4: No

---

## [Editor Report · Acceptance letter]

PONE-D-25-33803R1

PLOS ONE

Dear Dr. Han,

I'm pleased to inform you that your manuscript has been deemed suitable for publication in PLOS ONE. Congratulations! Your manuscript is now being handed over to our production team.

Kind regards,

on behalf of

Dr. Muhammad Salman Bashir

Academic Editor

PLOS ONE